# EVALUATION OF GENERATIVE NETWORKS THROUGH THEIR DATA AUGMENTATION CAPACITY

## ABSTRACT

Generative networks are known to be difficult to assess. Recent works on generative models, especially on generative adversarial networks, produce nice samples of varied categories of images. But the validation of their quality is highly dependent on the method used. A good generator should generate data which contain meaningful and varied information and that fit the distribution of a dataset. This paper presents a new method to assess a generator. Our approach is based on training a classifier with a mixture of real and generated samples. We train a generative model over a labeled training set, then we use this generative model to sample new data points that we mix with the original training data. This mixture of real and generated data is thus used to train a classifier which is afterwards tested on a given labeled test dataset. We compare this result with the score of the same classifier trained on the real training data mixed with noise. By computing the classifier's accuracy with different ratios of samples from both distributions (real and generated) we are able to estimate if the generator successfully fits and is able to generalize the distribution of the dataset. Our experiments compare different generators from the VAE and GAN framework on MNIST and fashion MNIST dataset.

## 1 INTRODUCTION AND MOTIVATION

Generative network approaches have been widely used to generate samples in recent years. Methods such as GAN (Goodfellow et al., 2014), WGAN (Arjovsky et al., 2017), CGAN (Mirza & Osindero, 2014), CVAE (Sohn et al., 2015) and VAE (Kingma & Welling, 2014) have produced nice samples on various image datasets such as MNIST, bedrooms (Radford et al., 2015) or imageNet (Nguyen et al., 2017).

One commonly accepted tool to evaluate a generative model trained on images is visual assessment to validate the realistic character of samples. One case of this method is called 'visual Turing tests', in which samples are visualized by humans who try to guess if the images are generated or not. It has been used to assess generative models of images from ImageNet (Denton et al., 2015) and also on digit images (Lake et al., 2015). Salimans et al. (2016) proposes to automate this method with the inception score, which replaces the human judgment by the use of a pretrained classifier to assess the variability of the samples with an entropy measure of the predicted classes and the confidence in the prediction. Unfortunately, those two methods do not indicate if the generator collapses to a particular mode of the data distribution. Log-likelihood based evaluation metrics were widely used to evaluate generative models but as shown in Lucas Theis & Bethge (2016), those evaluations can be misleading in high dimensional cases.

The solution we propose to estimate both sample quality and global fit of the data distribution is to incorporate generated data into the training phase of a classifier before evaluating it. Using generated samples for training has several advantages over using only the original dataset. First, it can make training more efficient when the amount of data is low. As shown in Ng & Jordan (2002), where the conditional distribution $P(Y|X)$(X represents the samples and Y the classes) learned by a generative model is compared to the same conditional distribution learned by a discriminative model, the generative model performs better in learning this conditional distribution by regularizing the model when the amount of data is low. Secondly, once the generative model is trained, it can

sample as much images as needed and can produce interpolations between two images which will induce less risk of overfitting on the data. Other works use generative models for data augmentation (Ratner et al., 2017) or to produce labeled data (Sixt et al., 2016) in order to improve the training of discriminative models, but their intention is not to use it to evaluate or compare generative neural networks.

Our method evaluates the quality of a generative model by assessing its capacity to fit the real distribution of the data. For this purpose, we use the samples generated by a given trained generative model. Our work aims to show how this data augmentation can benefit the training of a classifier and how we can use this benefit as an evaluation tool in order to assess a generative model. This method evaluates whether the information of the original distribution is still present in the generated data and whether the generator is able to produce new samples that are eventually close to unseen data. We compare classifiers trained over mixtures of generated and real data with varying ratios and with varying total amounts of data. This allows us to compare generative models in various data settings (i.e., when there is few or many data points).

The next section will present the related work on generative models, the exploitation of the generated samples and their evaluation. We then present our generative model evaluation framework before presenting experimental results on several generative models with different datasets.

## 2 RELATED WORK

### 2.1 GENERATIVE MODELS

The variational auto-encoder (VAE) framework (Kingma & Welling, 2014), (Rezende et al., 2014) is a particular kind of auto-encoder which has control over its latent space, in which each variable is a sample from a prior distribution, often chosen as an univariate normal distribution $\mathbf{N}(0, I)$ (where $I$ is the identity matrix). The VAE learns to map this low dimensional latent space to the observation space. This characteristic makes the VAE an interesting option for generating new data after training. The particularity of the latent space comes from the minimization of the KL divergence between the distribution of the latent space and the prior $\mathbf{N}(0, I)$. For the sake of simplicity, in this paper we will speak about the decoder of the VAE as a generator.

Generative adversarial networks (Goodfellow et al., 2014) are a framework of models that learn by a game between two networks: a generator that learns to produce images from a distribution $P$ and a discriminator which learns to discriminate between generated and true images. The generator wants to fool the discriminator and the discriminator wants to beat the generator. This class of generative models can produce visually realistic samples from diverse datasets but they suffer from instabilities in their training. Some recent approaches such as Wasserstein GAN (WGAN) (Arjovsky et al., 2017) try to address those issues by enforcing a Lipschitz constraint on the discriminator.

Conditional neural networks (Sohn et al., 2015) and in particular Conditional Variational Autoencoders (CVAE) or Conditional Generative adversarial networks (CGAN) (Mirza & Osindero, 2014) are a class of generative models that have control over the sample's class. By imposing a label during training, a conditional generative network can generate from any class and thus produces labeled data. The conditional approach has been used to improve the quality of generative networks and make them more discriminative (Odena et al., 2017). They are particularly adapted for our setup because we need to generate labeled data to train our classifiers.

### 2.2 EXPLOITATION AND EVALUATION OF GENERATED SAMPLES

In Ratner et al. (2017), a generator is used to perform data augmentation. Instead of designing a composition of fine tuned transformations for this objective, the authors use adversarial training to learn a sequence of incremental operations (for example rotating or swapping words in a sentence). Their approach uses a GAN to be able to generalize in terms of better data-augmentation and to increase their performance on different datasets such as Cifar10 and the ACE relation extraction task. Sixt et al. (2016) also learns a sequence of transformations with generative networks from the GAN family, but they use a 3D model as input and create an augmented view of it. Our approach is similar by using generative networks for data-augmentation but we do not attempt to learn transformations.

Instead, we use the generated data to assess if the generative model has been able to generalize over the distribution of the data.

The evaluation of generative networks is discussed in Lucas Theis & Bethge (2016). The authors show that different metrics (as Parzen windows, Nearest Neighbor or Log likelihood) applied to generative models can lead to different results. Good results in one application of a generative model can not be used as evidence of good performance in another application. Their conclusion is that evaluation based on sample visual quality is a bad indicator for the entropy of samples. Conversely, the log-likelihood can be used to produce samples with high entropy but does not assure good visual quality. The method we propose can both estimate the quality and the entropy of samples as we will show in Section 3.

The quality of the internal representation of a generator can also be estimated with a discriminator. In Radford et al. (2015) they use the discriminator of a ACGAN as feature extractor for evaluating the quality of unsupervised representation learning algorithms. They apply the feature extractor on supervised datasets and evaluate the performance of linear models fitted on top of these features. They experiment a good accuracy on Cifar10 thanks to this method. This approach gives insight on how the discriminator estimates if an image is true or false. If the discriminator has good enough feature extractors for classification, it means that the generator samples are hard to be discriminated from samples from the true distribution. It assess indirectly the quality of the generator. This method is however applicable only if a deep convolutional neural networks is used as discriminator and can not be applied, e.g., on variational auto-encoders. The principal difference between a discriminator and our classifier is that it is not involved in the training process. In our approach, the generator is completely independent from the classifier and therefore there is no bias from the classifier in the generator.

Parzen windows estimate is a method to estimate the unknown probability density function $f$ of a probability distribution $P$. This method uses a mixture of simpler probability density functions, called kernels, as approximates for $f$. In general, a popular kernel used is an isotropic Gaussian centered on a given data point with a small variance (the variance is an hyper parameter here). The idea, like other methods based on Kernel Density Estimation, is to have a small window on each data point such that we apply some smoothing over the function we try to approximate. However, even if the number of samples is high, Parzen windows estimator can be still very far from the true likelihood as shown in Lucas Theis & Bethge (2016), and thus cannot be a good approach to evaluate if the data distribution learned by a model is close to the original one.

Multi-scale structural similarity (MS-SIM, Wang et al. (2003)) is a measurement that gives a way to incorporate image details at different resolutions in order to compare two images. This similarity is generally used in the context of image compression to compare image before and after compression. In Odena et al. (2017) the authors use this similarity to estimate the variability inside a class. They randomly sample two images of a certain class and measure the MS-SIM. If the result is high, then images are considered different. By operating this process several times, the similarity should give an insight on the entropy of $P(X|Y)$ (X a data point and Y its class): if the MS-SIM gives high result, the entropy is high; otherwise, the entropy is low. However, it can not estimate if the sample comes from one or several modes of the distribution $P(X|Y)$. For example, if we want to generate images of cats, the MS-SIM similarity can not differentiate a generator that produces different kinds of black cats from a network that produces different cats of different colors. In our method, if the generator is able to generate in only one mode of $P(X|Y)$, the score will be low in the testing phase.

Another approach that aims to evaluate a generative model by using a conditional distribution learned by a classifier is the inception score (Salimans et al., 2016; Odena et al., 2017). The authors use a pretrained inception classifier model to get the conditional label distribution $P(Y|X)$ over the generated samples. They proposed the following score in order to evaluate a generative model:

$$\exp(\mathbb{E}_X KL(P_{data}(Y|X) \parallel P(Y))), \qquad (1)$$

When the score is high, the generator produces samples on varied classes (Cross entropy of $P(Y|X), P(Y)$ is high) and the samples look like real images from the original dataset (entropy of $P(Y|X)$ is low ). Inception score can be seen as a measurement of the variability of the generated data while penalizing the uncertainty of $P(Y|X)$. Unfortunately, it does not estimate if the samples are varied inside a certain class (the entropy of $p(X|Y)$). Our approach imposes a high entropy of $P(Y)$ and gives an unbiased indicator about the entropy of both $P(Y|X)$ and $P(X|Y)$.

## 3 METHODS

We evaluate generators in a supervised training setup. We have a dataset $D$ composed of pair of examples $(x, y)$ where $x$ is a data point and $y$ the label associated to this data point. The dataset is split in three parts $D_{train}, D_{valid}$ and $D_{test}$. Our method needs a generative model that can sample conditionally from any given label $y$. This conditional generative model is thus trained on $D_{train}$. Once the training of this model is done, we sample random labels and use the generative model to get a new dataset $D_{gen}$. Then, we mix the true training set $D_{train}$ with the new generated data $D_{gen}$. The ratio of generated data into the whole mixture $D_{mix}$ is called $\tau$. $\tau$ is used as the probability to sample a generated batch rather than a batch of true data during training. $D_{mix}$ is used to train a classifier on the classes of the dataset. This classifier is evaluated at each epoch over the portion $D_{valid}$ of the dataset. Once, we get the best model over $D_{valid}$, we compute the score of this classifier over the test set $D_{test}$. We compare the results from training on $D_{mix}$ with a baseline. The baseline is the score of the same classifier model trained only on training data $D_{train}$. Our conditional generative model evaluation can be seen as a ratio between those two scores. We can summarize our method as follows:

1. Train a conditional generative model over $n$ sample of $D_{train}$
2. Mix the samples $D_{gen}$ generated by this model with $D_{train}$ under a probability $\tau$ into $D_{mix}$
3. Train a discriminative model over $D_{mix}$
4. Train a discriminative model over $D_{train}$ using the $n$ same sample as for the generator training
5. Select a classifiers over a valid set $D_{valid}$.
6. Compare the score of those classifiers over a test set $D_{test}$.

By iterating this method on diverse values of $\tau$ and $n$ we can evaluate the quality of a generator given a dataset.

## 4 EXPERIMENTS

### 4.1 EXPERIMENTAL PROTOCOL

Often, generative models are presented on popular datasets like MNIST. Fashion-MNIST (Xiao et al., 2017) can also serve as a direct drop-in replacement for the original MNIST dataset. This dataset is however more complex than MNIST as images have a higher variability. Thus, we use these datasets in order to evaluate different generative models.

We used two different methods in order to get conditional generative models. The first uses traditional generative neural network which can not produce labeled data. In order to associate each generated sample to a label, we train one generator for each specific class $y$ on $D_{train}$. This makes us able to label any generated sample. Once the training of those generators is done, we mix the samples obtained by each generator in order to produce $D_{gen}$. For the experiments, we compare two generative models is this setting: a standard VAE and a WGAN. The second method uses conditional generative models which can generate samples in all classes while controlling the class of a particular sample. Conditional models can thus generate various labeled samples and produce a whole dataset $D_{gen}$ directly. In this last case, we ran our experiments on CVAE and CGAN. Once the dataset $D_{gen}$ is generated, we mixed it with the real dataset $D_{train}$. As we can generate as much data as we want, we experimented different ratios between real datasets and generated datasets. We call $\tau$ the probability of sampling from $D_{gen}$. We made experiments with different values for $\tau = [0.000, 0.125, 0.250, 0.375, 0.500, 0.625, 0.750, 0.875, 1.000]$. $\tau = 0$ implies that we use only data from $D_{train}$. In this specific setting, we compare the effectiveness of the data augmentation with generated samples versus classic data augmentation as isotropic Gaussian noise with an optimized variance or a random pixel dropout with a probability $\alpha$ of putting a pixel to 0. We also train a classifier without any data augmentation as baseline.

We use a standard CNN with a softmax output as classifier to predict the labels on this mixture of samples. On each epoch we evaluate this classifier over a validation set $D_{valid}$. Then, we choose

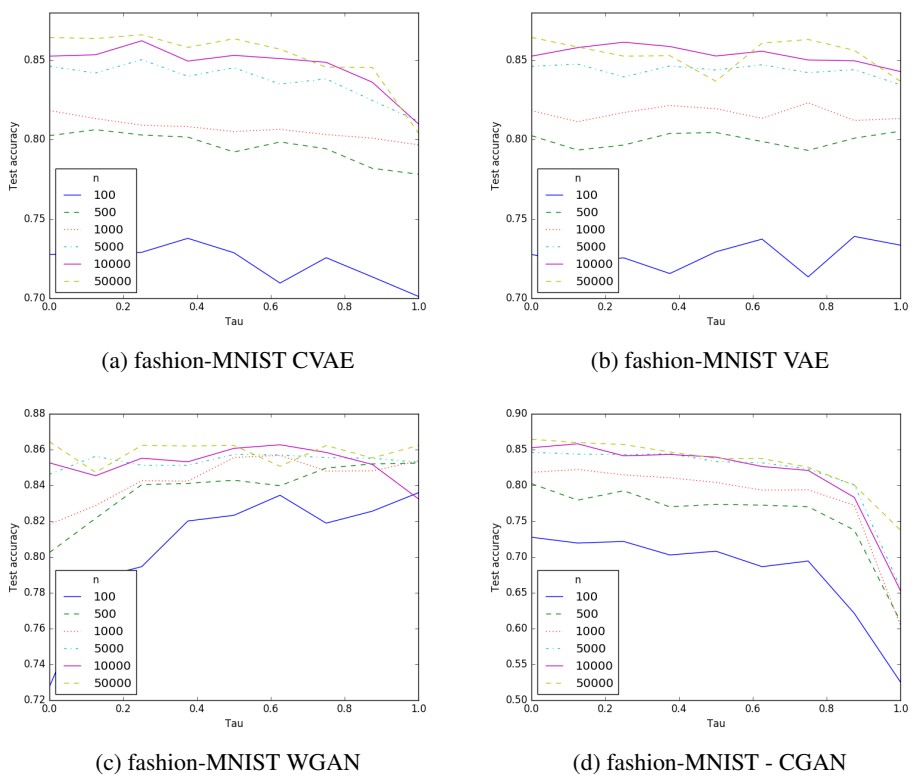

Figure 1: Representation of test accuracy when $\tau$ increases on fashion-MNIST (MNIST in appendix) with different amounts of training data.

the classifier that performs best on this validation set. We use early stopping to stop the training if the accuracy does not improve anymore on the validation set after $50$ epochs. The classifier is then tested on $D_{test}$. We train a classifier for each value of $\tau$. We assess the quality of the generative model by comparing the test score of this classifier when $\tau = 0$ versus the best test score of the classifier with $\tau > 0$. The result gives an indication on how the learned distribution from $D_{train}$ fits and generalizes the distribution from $D_{train}$. In order to be able to compare results from different generators on a given dataset, we always use the same classifier architecture.

To be able to estimate the impact of learning from generative settings versus discriminative ones, we have made variable the amount of data in $D_{train}$ used to train both generator and classifier. Thus, we repeat all our experiments for the following amount of data samples: $[100, 500, 1000, 5000, 10000, 50000]$. This allows us to measure the regularization capacity of the generated samples over the classifier's training. We interpret this regularization capacity of those samples as a capacity of generalization.

## 4.2 RESULTS

In Figure 1, we present the test accuracy when $\tau$ increase. When $\tau = 0$ there is no generated data, this is the result of the baseline without data augmentation. Our interpretation of the figure is that if the accuracy is better than baseline with a low $\tau$ ( $< 0.5$) it means that the generator is able to generalize by learning meaningful informations about the dataset. When $\tau > 0.5$ if the accuracy is maintained it means the generated data can replace the dataset in most parts of the distribution. When $\tau = 1$ there is no more original data, the classifier is thus trained only on generated samples. If the accuracy is still better than the baseline, it means that the generator has fit the training distribution (and eventually has learned to generalize if this score is high over the test set).

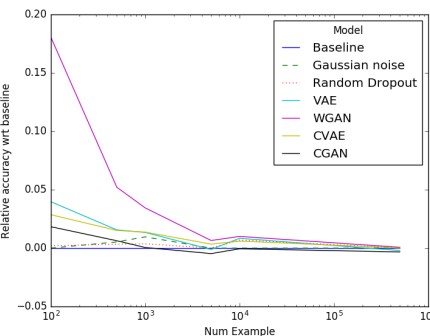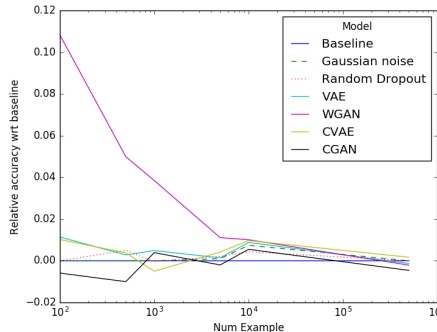

(a) Relative accuracy wrt. baseline on mnist for different models

(b) Relative accuracy wrt. baseline on fashion-mnist for different models

Figure 2: Representation of the data augmentation capacity of each generative models. For each number of training example we show the maximum accuracy a generative model can achieve by tuning $\tau$. We also show results when tuning hyper-parameter for data augmentation method.

Following this interpretation, Figure 1 allows us to compare different generative neural networks on fashion-MNIST. For example we can see that VAE (Figure 1b) and CVAE (Figure 1a) are able to maintain the baseline accuracy when we do data augmentation with generated samples for different amount of data $n$. But the accuracy goes down when there are only generated data, which means that these generative models in those settings did not fit the whole distribution. CGAN has the same kind of behavior but the degradation is worse when $\tau = 1$. However WGAN is able improve the accuracy for $\tau < 0.5$ and to maintain it even when $\tau = 1$. Figure 1 also allows to estimate the entropy of $P(X)$. As a deep neural network can not be trained efficiently without various data samples, if the the accuracy on test set is good with only generated data ($\tau = 0$), necessary the entropy of $P(X)$ is high as the entropy of $P(X|Y)$. This condition is sufficient but not necessary to assess the quality of a generator on a given task.

Our result shows that training one WGAN per class is the best solution to fit the complete distribution, whatever the number of training data used. The bad results on CGAN can be explained by the difficulties to train this model and it's instability.

Figure 2 represents the best accuracy that each model can achieve when $\tau$ is tuned. It shows that, aside from CGAN on MNIST, all generator can be used to increase accuracy, whatever the number of training data. In this context $\tau$ can be seen as a tunable hyper parameter for data augmentation. Figure 2 also shows that the capacity of generalization is particularly effective when the number of example is low. This can be explained by the fact that when the data number increases the need of data augmentation decrease.

$$\Psi_G = \frac{1}{n} \sum_n max_\tau [(acc_n(G, \tau))] - acc_n(\tau = 0) \, , \tag{2}$$

The results shown in Table 1 summarize Figure 2 for each generator $G$ by a numerical value $\Psi_G$ (Eq. 2). We call $\Psi_G$ the data augmentation capacity of a generator. It is computed, for a given generator $G$, by the mean of the differences between the accuracy on mixture with $\tau$ tuned and the baseline accuracy for each number $n$ of training data. The different number of data is arbitrary chosen. In our experiment we compute it with $[0.2\%, 1\%, 2\%, 10\%, 20\%, 100\%,]$ of the train set of the dataset. The important thing is to operate at different scales to estimate how the generative model is able to generalize.

The results of Table 1 indicates if a generative model is globally able to perform data augmentation for a given data set. A positive result indicates that the generative model is able to generalize well on the dataset at different sizes. In case of low amount of data, it is better to refer to Figure 2 to choose the best model for that specific case.

Table 1: Results Table : $\Psi_G$ values

| Datasets | Gaussian | Random Dropout | VAE | CVAE | WGAN | CGAN |
|---|---|---|---|---|---|---|
| MNIST | 0.0024 | 0.0027 | 0.0123 | 0.0116 | 0.0474 | 0.0027 |
| Fashion MNIST | 0.0014 | 0.0018 | 0.0047 | 0.0041 | 0.0357 | -0.0021 |

## 5 Discussion

We presented a method to estimate how well a generative model has learned to generalize in a conditional setting. Using generated samples as data augmentation in order to improve a discriminative model is a well known technique. However, assessing the quality of a generative model with a discriminative model seems to have been less explored. As we have shown, this evaluation is meaningful to measure how well the generative model can sample data points that are probable under the true distribution we want to approximate. In this paper we applied this method on images samples. It means that we can correlate our measure with a visually assess of the samples quality as the generator outputs are in pixel space. Our assessment method can also be used in other spaces as long as labeled data are available.

The relative benefits of discriminative and generative models have been studied in Ng & Jordan (2002). They found that for a small number of training examples $n$, a generative model will be less prone to overfitting. A discriminative model on a small number of examples risks to learn some spurious nodes that will penalize the generalization capacity. Our results are coherent with their conclusion as the data augmentation induced by generated data is particularly effective when $n$ is low.

Our evaluation was performed on several datasets, generative models, different ratios and amounts of training data. With the current results, WGAN seems to be the most efficient solution. However, this result should be confirmed by experiments on other datasets, generative models and with different types of discriminative models to get a more general comparison. This will be explored in further experiments.

As presented in White (2016), the sampling method of a generator can be adapted, which can have an impact on the data produced. A way to boost the performance of a generator can be to focus on improving the sampling phase instead of the model design or the training. An extension of this work could be to look into the impact of several sampling techniques on the performance of a generative model.

## 6 Conclusion

This paper introduces a new method to assess and compare the performances of generative models on various labeled datasets. By training a classifier on several mixture of generated and real data we can estimate the ability of a generative model to generalize. When addition of generated data into the training set achieved better data augmentation than traditional data augmentation as Gaussian noise or random dropout, it demonstrates the ability of generative models to create meaningful samples. By varying the number of training data, we compute a data augmentation capacity $\Psi_G$ for each model on MNIST and fashion-MNIST datasets. $\Psi_G$ is a global estimation of the generalization capacity of a generative model on a given dataset. The results presented here are produced on image datasets but this method can be used on all kinds of datasets or generative models as long as labeled data is available.

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

# A ADDITIONAL RESULTS

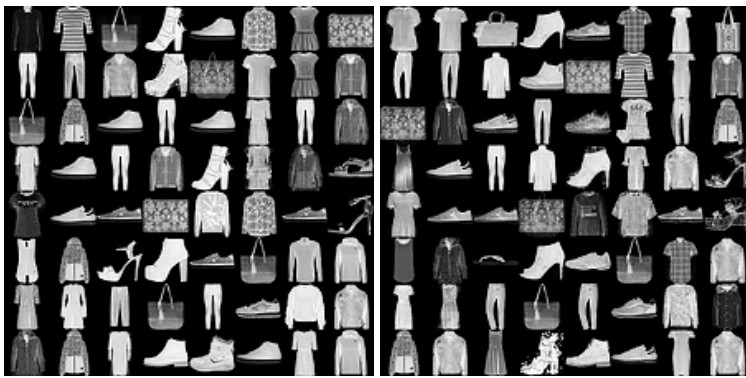

(a) VAE samples when trained with 50 images

(b) VAE samples when trained with 100 images

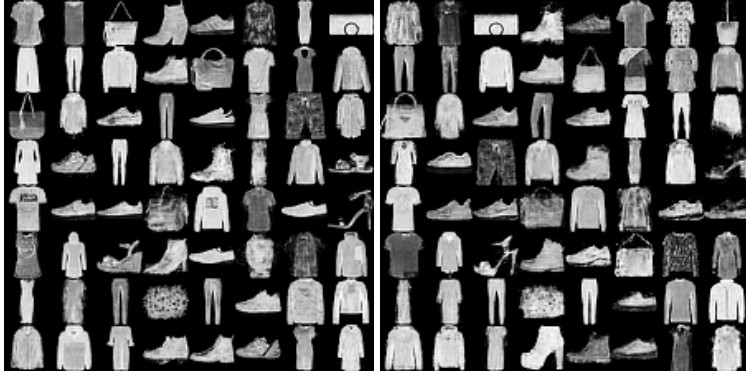

(c) VAE samples when trained with 500 images

(d) VAE samples when trained with 1000 images

Figure 3: This figure presents samples from VAEs when trained with different amounts of data. It represents the overfitting capacity of a VAE. All four samples set look good, but for example, the top left trained with only 50 different data often produce similar images (as the samples on top right trained with 100 images). When the number of training images increases the variability seems good afterwards but as we can see in Figure 4c when $\tau = 1$ the generator generalizes better the distribution when n is $< 1000$ than when $> 1000$ is high.

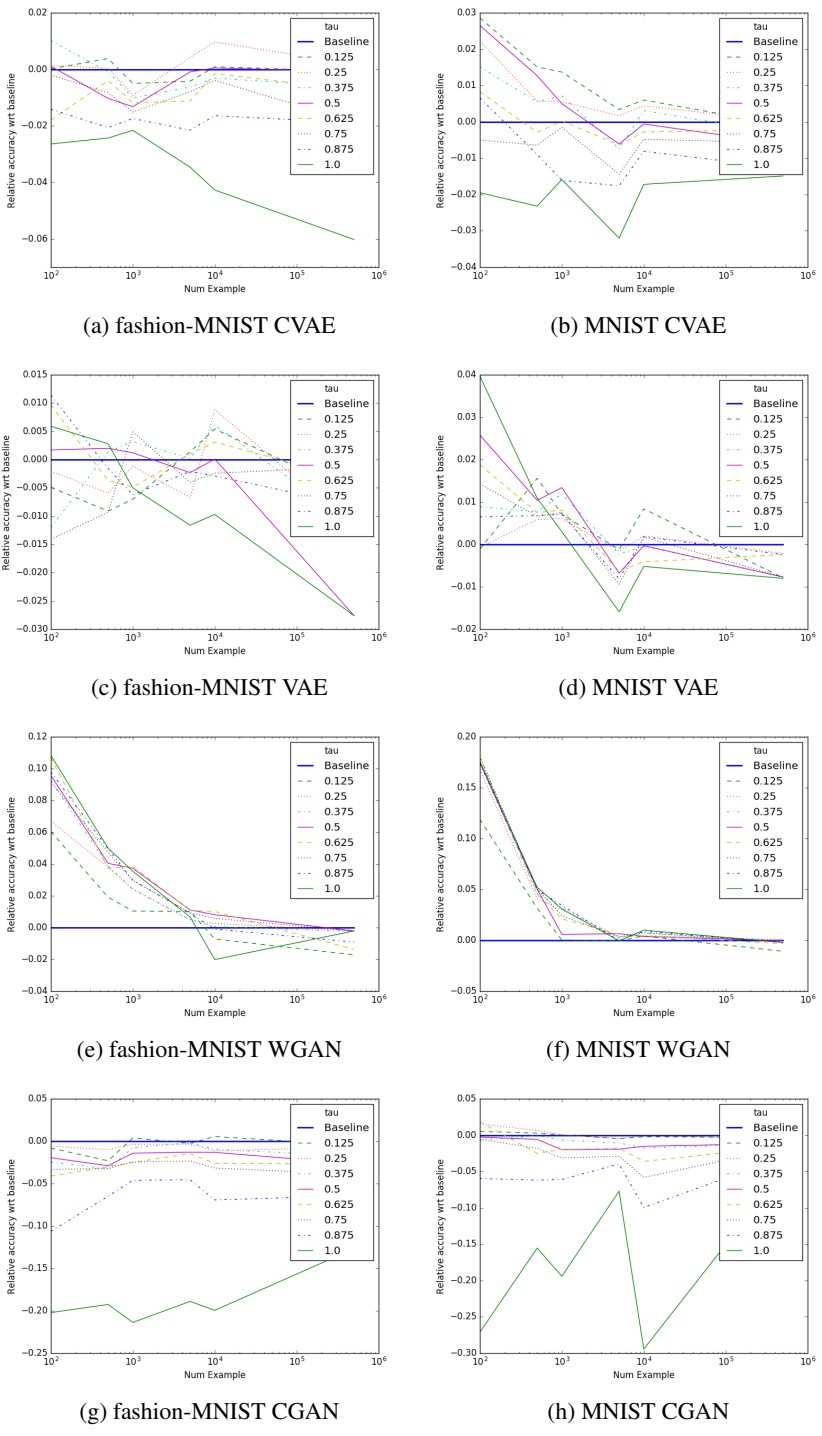

Figure 4: Relative accuracy improvement between the baseline trained on original data and the accuracy with generated or noise data augmentation in training. $\tau$ is the ratio between the number of generated data and the total number of data used for training.

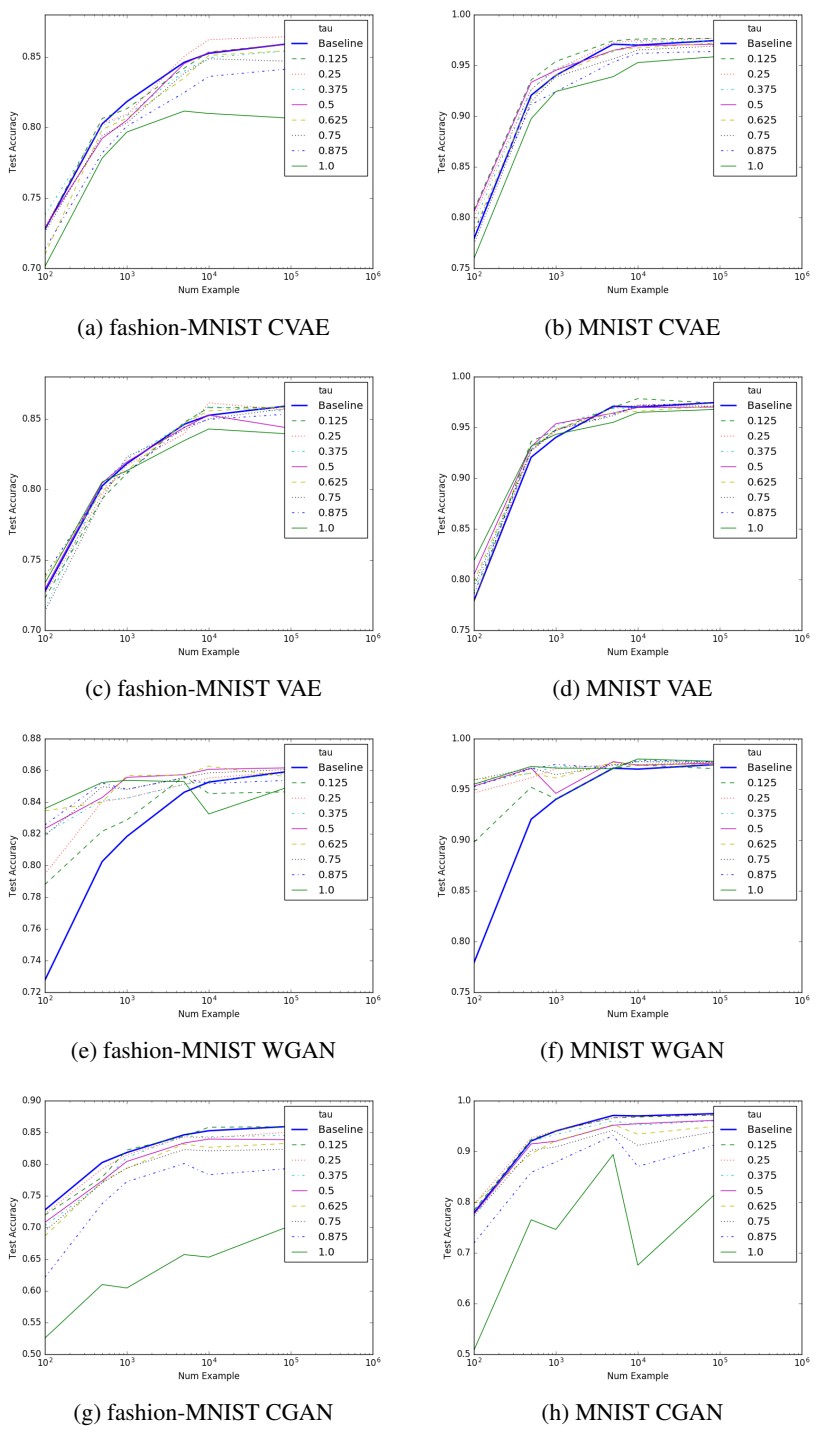

Figure 5: Test accuracy on each model for different values of $\tau$ and different amounts of training data.

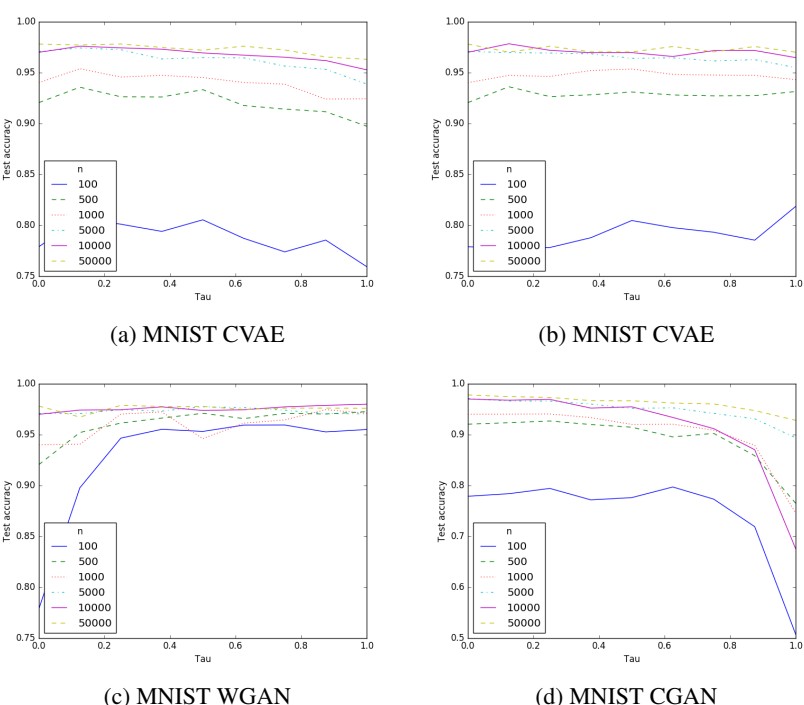

(a) MNIST CVAE

(b) MNIST CVAE

(c) MNIST WGAN

(d) MNIST CGAN

Figure 6: Test accuracy when $\tau$ increases with different amounts of training data on MNIST.

