# OpenReview forum: "Evaluation of generative networks through their data augmentation capacity"
_ICLR.cc/2018/Conference — Reject_

### Official Review · AnonReviewer3 · 2017-11-27
**insufficient evaluation**

**Rating:** 3
**Confidence:** 5

**Review:**

The main idea is to use the accuracy of a classifier trained on synthetic training examples produced by a generative model to define an evaluation metric for the generative model. Specifically, compare the accuracy of a classifier trained on a noise-perturbed version of the real dataset to that of a classifier trained on a mix of real data and synthetic data generated by the model being evaluated. Results are shown on MNIST and Fashion MNIST.

The paper should discuss the assumptions needed for classifier accuracy to be a good proxy for the quality of a generative model that generated the classifier's training data. It may be the case that even a "bad" generative model (according to some other metric) can still result in a classifier that produces reasonable test accuracy. Since a classifier can be a highly nonlinear function, it can potentially ignore many aspects of its input distribution such that even poor approximations (as measured by, say, KL) lead to similar test accuracy as good approximations.

The sensitivity of the evaluation metric defined in equation 2 to the choice of hyperparameters of the classifier and the metric itself (e.g., alpha) is not evaluated. Is it possible that a different choice of hyperparameters can change the model ranking? Should the hyperparameters be tuned separately for each generative model being evaluated?

The intuition behind comparing against a classifier trained on a noise-perturbed version of the data is not explained clearly. Why not compare a classifier trained on only (unperturbed) real data to a classifier trained on both real and synthetic data?

Evaluation on two datasets is not sufficient to provide insight into whether the proposed metric is useful. Other datasets such as ImageNet, Cifar10/100, Celeb A, etc., should also be included.

---

> ### Author Response · Authors · 2017-12-22
> **Answer**
>
> Thanks for the review,
>
> Is the classifier accuracy a good proxy ? The classifier we use is a deep net ( 2 convolutional layer, one dropout, 2 fully connected). We don’t have mathematical proof but the idea is that if the (generated) training data is biased with respect to the (real) testing data, the test error will be large. So being able to train or improve training with generated data empirically indicates that the “fake” data are part of the same manifold than the testing data and cover most of this manifold. Therefore we can assume that the generated data have high variability and good (enough) visual quality.
> However for some classifiers, the classification accuracy would not be representative as KNN. We were more thinking of deep net classifier which are harder to train successfully without good training data.
> However for some classifiers, the classification accuracy would not be as representative. For example, KNN could have good accuracy by taking advantage of a few good samples while  ignoring bad samples. On the contrary, CNN are trained to be able to create representations from all training data and use them for classification. Bad training data will induce learning bad representations and usually bad generalization in classification.
>
> The hyperparameters can indeed change the ranking, like in any other classification algorithm, they have to be tuned to assess a particular generative model with a particular dataset in order to reach the best possible performance. We did not had time to evaluate the sensitivity of the equation 2.
>
> - Why not compare a classifier trained on only (unperturbed) real data?
> We did it as, in Figure 1, it corresponds to tau=0. But this comparison against real data is unfair, in the sense that when we have samples from generative model we add some noise, the model will never see twice the same sample. And it’s known that classifiers are more robust when training with perturbed data.
>
> - comparison with noise data augmentation
> The reason to compare data augmentation (DA) from generative model with classic DA methods was to show that the generative model produce better DA than just random perturbation. It also gives insight on how the metric evaluate simple data augmentation. Therefore the DA introduced by generative models is not only due to a bad reconstruction that would introduce variability in the training data.
>
> - comparison with other classifier / dataset
> We plan to test other classifier and other dataset to compare the performance. However, making several generative model work on the same dataset is not an easy task. We plan to use LSUN and tiny-imagenet for further experimentation.

---

### Official Review · AnonReviewer2 · 2017-11-28
**Not very novel, weak analysis and justification.**

**Rating:** 3
**Confidence:** 5

**Review:**

The authors propose to evaluate how well generative models fit the training set by analysing their data augmentation capacity, namely the benefit brought by training classifiers on mixtures of real/generated data, compared to training on real data only. Despite the the idea of exploiting generative models to perform data augmentation is interesting, using it as an evaluation metric does not constitute an innovative enough contribution.

In addition, there is a fundamental matter which the paper does not address: when evaluating a generative model, one should always ask himself what purpose the data is generated for. If the aim is to have realistic samples, a visual turing test is probably the best metric. If instead the purpose is to exploit the generated data for classification, well, in this case an evaluation of the impact of artificial data over training is a good option.

PROS:
The idea is interesting.

CONS:
1. The authors did not relate the proposed evaluation metric to other metrics cited (e.g., the inception score, or a visual turing test, as discussed in the introduction). It would be interesting to understand how the different metrics relate. Moreover, the new metric is introduced with the following motivation “[visual Turing test and Inception Score] do not indicate if the generator collapses to a particular mode of the data distribution”. The mode collapse issue is never discussed elsewhere in the paper.

2. Only two datasets were considered, both extremely simple: generating MNIST digits is nearly a toy task nowadays. Different works on GANs make use of CIFAR-10 and SVHN, since they entail more variability: those two could be a good start.

3. The authors should clarify if the method is specifically designed for GANs and VAEs. If not, section 2.1 should contain several other works (as in Table 1).

4. One of the main statements of the paper “Our approach imposes a high entropy on P(Y) and gives unbiased indicator about entropy of both P(Y|X) and P(X|Y)” is never proved, nor discussed.

5. Equation 2 (the proposed metric) is not convincing: taking the maximum over tau implies training many models with different fractions of generated data, which is expensive. Further, how many tau’s one should evaluate? In order to evaluate a generative model one should test on the generated data only (tau=1) I believe. In the worst case, the generator experiences mode collapse and performs badly. Differently, it can memorize the training data and performs as good as the baseline model. If it does actual data augmentation, it should perform better.

6. The protocol of section 3 looks inconsistent with the aim of the work, which is to evaluate data augmentation capability of generative models. In fact, the limit of training with a fixed dataset is that the model ‘sees’ the data multiple times across epochs with the risk of memorizing. In the proposed protocol, the model ‘sees’ the generated data D_gen (which is fixed before training) multiple time across epochs. This clearly does not allow to fully evaluate the capability of the generative model to generate newer and newer samples with significant variability.


Minor:
Section 2.2 might be more readable it divided in two (exploitation and evaluation).

---

> ### Author Response · Authors · 2017-12-22
> **Answer**
>
> Thanks for your review.
>
> We assumed that the purpose of the GAN is to be able to fit the distribution of a given dataset, not only to be able to generate some nice realistic samples.
>
> 1 . the inception score or a visual turing test alone only address the realistic characteristic of sample not their variability. It could be interesting to compare but it is easy to make a model overfit to make inception and visual turing test good while our method would detect the overfit.
>
> 2 . We agree that we used toy example. We experimented cifar10 but did not add the results because we did not achieve to make all the generative models work on it. Some papers present result in cifar10 but the training is very hard to design (looking at the sample is clearly enough to know when a training absolutely does not work). We are planning to also use LSUN and tiny-imagenet.
>
> 3 . As the method is based on data only (generate and true), it is designed for any type of generative models. We took GANs and VAEs as example, the goal was to present the idea, but we are not able to experiment all possible generative models in a reasonable amount of time. However, we would be interested by suggestions of other models to compare.
>
> 4 . We indeed did not proved it (and we will rephrase the paragraph to explain it is the intuition behind the approach) but we impose high entropy of P(Y) because we sample uniformly the different classes. The indicator is unbiased because it's evaluated on never seen data. We evaluate the entropy of P(X|Y) and P(Y|X) because we need a good variability of the data with not too much uncertainty in P(Y|X) for each class to have a good training of the classifier (we could add the result class by class).
>
> 5 . Tau=1 is a difficult setting as it necessitates to be able to fit the whole training distribution. We wanted to add simpler settings where the generative model can show that even if it is not able to fit the whole distribution, it can generalize to some extend. When tau is considered as an hyperparameter, it leave the possibility to Generative model developers to choose a particular value to highlight some behavior of the mode. Besides this, finding the best tau is expensive but assessing visual data variability is a difficult problem.
>
> 6 . It's not well explained in the paper, we will update it, but D_gen is generated so that the classifier only uses newly generated samples and none of the generated sample is used multiple times.

---

### Official Review · AnonReviewer1 · 2017-11-28
**Potentially useful technique for evaluation of generative models (need experiments on real tasks)**

**Rating:** 5
**Confidence:** 3

**Review:**

The paper proposes a technique for analysis of generative models.

The main idea is to (1) define a classification task on the underlying data, (2) use the generative model to produce a training set for this classification task, and (3) compare performance on the classification task when training on generated and real training data.

Note that in step (2) is it required to assign class labels to the generated samples. In this paper this is achieved by learning a separate generative model for each class label.

Summary:

I think the proposed technique is useful, but needs to be combined with other techniques to exclude the possibility that model just memorized the training set. To be stronger the paper needs to consider other more realistic tasks from the literature and directly compare to other evaluation protocols.

Positive:
+ the technique operates directly on the samples from the model. It is not required to compute the likelihood of the test set as for example is needed in  the "perplexity" measure). This makes the technique applicable for evaluation of a wider class of techniques.

+ I like the result in Fig. 1. There is a clear difference between results by WGAN and by other models. This experiment convinces me that the peroposed analysis by augmentation is a valuable tool.

+ I think the technique is particularly valuable verify that samples are capturing variety of modes in the data. Verifying this via visual inspection is difficult.

Negative:

- I think this metric can be manipulated by memorizing training data, isn't it? The model that reproduces the training set will still achieve good performance at \tau = 1, and the model that does simple augmentation like small shifts / rotations / scale and contrast changes might even improve over training data alone. So the good performance on the proposed task does not mean that the model generalized over the training dataset.

- I believe that for tasks such as image generating none of the existing models generate samples that would be realistic enough to help in classification. Still some methods produce images that are more realistic than others. I am not sure if the proposed evalaution protocol would be useful for this type of tasks.

- The paper does not propose an actual metric. Is the metric for performance of generative model defined by the best relative improvement over baseline after tuning \tau as in Tab. 1?  Wouldn't it be better to fix \tau, e.g. \tau = 1?

- Other datasets / methods and comparison to other metrics. This is perhaps the biggest limitation for me right now. To establish a new comparison method the paper needs to demonstrate it on relevant tasks (e.g. image generation?), and compare to existing metrics (e.g. "visual inspection" and "average log-likelihood").

---

> ### Author Response · Authors · 2017-12-22
> **Answer**
>
> Thanks for your review.
>
> Using just memorization should normally give a result of 0 (which is not so bad). For a given number of example, a result better than 0 indicates that the generative model is able to achieve data augmentation because it leads to better performance than the baseline. ‘Traditional’ data augmentation can also be compared to DA from generative models. We only included gaussian noise and random dropout to give a simple comparison but our first goal is to compare generative models.  It also gives insight on how the metric evaluate simple data augmentation.
>
> Knowing if generative models can really help for classification is not the goal of the paper. The result we provide gives case where it works (in simple setting) but the important thing here is that the ‘metric’ makes it possible to discriminate between generative models. If it’s with negative results, it is still valid.
>
> As you suggest, comparing with alternatives is obviously important and we will compare with other metrics for a next submission.

---

### Decision · Program_Chairs · 2018-01-29
**ICLR 2018 Conference Acceptance Decision**

**Decision:**

Reject

**Comment:**

Given that the paper proposes a new evaluation scheme for generative models, I agree with the reviewers that it is essential that the paper compare with existing metrics (even if they are imperfect). The choice of datasets was very limited as well, given the nature of the paper. I acknowledge that the authors took care to respond in detail to each of the reviews.